# Gamifying App-Based Low-Intensity Psychological Interventions to Prevent Sports Injuries in Young Athletes: A Review and Some Guidelines

**DOI:** 10.3390/ijerph182412997

**Published:** 2021-12-09

**Authors:** Víctor J. Rubio, Aurelio Olmedilla

**Affiliations:** 1Department of Biological and Health Psychology, Universidad Autonoma Madrid, 28049 Madrid, Spain; victor.rubio@uam.es; 2Department of Personality, Assessment and Psychological Treatments, University of Murcia, 30100 Murcia, Spain

**Keywords:** sports injuries, psychological intervention, low-intensity interventions, app-based interventions, gamification

## Abstract

Sports injuries have become a real health concern. Particularly noticeable is the increasing number of severe sports injuries among young people. Sports injury (SI) is a multifactorial event where many internal and external, proximal and remote factors play a role in a recursive way, including physical and psychological variables. Accordingly, many voices expressing the need of tackling that and several prevention programs have arisen. Nevertheless, different barriers and limitations prevent a wide extension of well-controlled programs, closely monitored by highly specialized professionals in ordinary sports grass-root organizations. These have helped flourishing different low intensity (LI)-interventions and e-Health apps focusing on both physical warmup, training and fitness, and mental skills aimed at reducing athlete’s vulnerability to SIs. This kind of intervention usually uses self-administered techniques and/or non-specialized staff that can effectively monitoring the program. In fact, LI-interventions have shown to be effective coping with different health and psychological issues. However, these interventions face an important challenge: the lack of engagement people usually show. The current paper proposes how gamification can contribute to the engagement to such interventions. Based on the mechanics–dynamics–aesthetics framework to analyze game design, the paper suggests a set of guidelines app- and web-LI interventions aimed at preventing SIs should include to foster motivation and reduce attrition.

## 1. Introduction

Deterding et al. defined gamification as “the use of game design elements in non-game contexts” [1]. According to Kim and Lee [2], gamification is aimed at “encourage greater engagement in people and aiding in creating richer experiences in everyday life events through game mechanisms and most importantly, with more enjoyment”. Kim and Lee also posed that gamification has pierced the arena of learning devices and tools, whether teaching languages or math, promoting healthy lifestyles or fostering military training, simulating driving, or instructing work competences.

Gamification has also shown to be a promising approach to increase user engagement with e-Health apps [3]. For instance, Höchsmann et al. [4] found better levels of adherence to physical exercise when comparing the results of a gamified phone-based intervention with a one-time direct counselling on lifestyles. 

What do games (and, by extension, gamified interventions focused on learning skills and changing behavior) make a useful tool to foster learning skills and promote behavioral change? As Plass, Homer and Kinzer [5] have synthesized, there are different aspects involved in why games are effective in promoting learning. The common mentioned aspect is motivation. There is no learning without engagement with the learning task and games may play a crucial motivation function. Motivation is supposed to promote individual’s engagement. However, motivation is necessary but not sufficient [6]. Plass et al. [5,7], distinguish between affective engagement, behavioral engagement, and sociocultural engagement but them all should foster the cognitive engagement in order to reach the learning outcomes. Adaptivity is another aspect authors mentioned as key in effectively promoting learning through games. Personalizing the game to the specific level or situation the learner presents promotes engagement and increases learning outcomes. Adapting the task to the current participant’s skill level, emotional state or any other condition makes the app less susceptible from attrition [8].

## 2. Low-Intensity Interventions

A low intensity (LI) intervention regards to a kind of intervention which, at least partially, does not hinge on a highly qualified practitioner but use either para-professionals, self-help materials and techniques, and/or e-Health. Typically, low intensity interventions are simple and brief, emphasize the value of between sessions homework, and assess, monitor, and evaluate progress as an intrinsic part of the intervention [9].

LI interventions present strong advantages compared to the disadvantages they may face. Firstly, they increase the access to interventions that otherwise, individuals would not have received. Moreover, they reduce the time a specialist is used to spend with an individual client, whether caring for more than one user at the same time or shortening the length and/or number of sessions in which the expert is attending the clients. Furthermore, they use non-specialized workers, peer supporters, etc. who are much more accessible both from the point of view of the availability and the fee. Finally, they provide quicker access to early and preventive interventions, fairly before the concern has become more severe or chronic [9].

LI interventions deliver content in a variety of flexible forms which maximize the opportunity for patient choice. Ten years ago, it mainly relied on self-help manuals, phone, email and SMS communication though, currently, the pervasiveness of smartphones drove to a new impulse on such kind of approaches that has propelled e-Health. The universalization of these devices together with the possibility of assessing relevant variables and providing the exercises in real time or nearly the real time, immediately before and after the significant events and privately and at the convenience of the user made app-supported resources key in the development of LI-interventions [10,11]. 

## 3. LI-Interventions and e-Health Apps to Reduce Sport Injury Rates

Sports injuries are usually associated to professional and high-performance sport where results and preserving health is not always well-balanced [12]. Nevertheless, sports injuries have become a global pandemic, affecting any kind of practitioners, including children and adolescents, regardless categories [13].

### 3.1. Epidemiology of Sport Injuries among Youngsters

Epidemiological data show a remarkable risk of sustaining sports injuries among youngsters, particularly among widely practiced sports [14,15,16,17]. Most of these studies are following the Strengthening the Reporting of Observational Studies in Epidemiology (STROBE) to collect and analyze epidemiological data [18] which make them comparable. According to these guidelines, authors have computed incidence rate as the number of injuries sustained per 1000 h of sport participation [19]. In this line, Caine et al. [14] found rates of injury per 1000 h of exposure ranging as high as from 5.0 to 34.4 in ice hockey, from 3.4 to 13.3 in rugby, and from 2.3 to 7.9 in soccer, among boys, and from 2.5 to 10.6 in soccer, 3.6 to 4.1 in basketball, and 0.5 to 4.1 in gymnastics, among girls. A recent study in basketball showed even higher injury rate/1000 h: 13.8 (99.4% confidence interval (CI): 11.2–16.8) in females and 14.8 (99.4% CI: 11.7–18.88) in males [20]. Likewise, Hootman et al. [16] found college athletes injury rates ranging from 4 to 13.8/1000 h depending on sports. Hammer et al. [15] reported incidence rate up to 7.04/1000 h in all the sports analyzed as a whole. Prieto-Gonzalez et al. [17] have found lower average rates (2.64/1000 h) though soccer ratcheted up to 7.21/1000 h.

In recent years, we have seen the increase in competitiveness and the expectations regarding youth sport goals from media, parents, coaches, and athletes themselves that, together with the rise in the number of practitioners, might lay at the basis of the current figures of sport injuries among young people [13,21,22]. Such factors, among others, may have provoked a bulge in mental and physical load which gears up the risk of injuries. In fact, recent studies have shown the relationships between physical and mental load and SI, particularly when young athletes assume training intensification [23].

### 3.2. LI-Interventions and e-Health Apps to Reduce Sports Injury Rate

The magnitude of the sports injury incidence rate has raised many voices expressing the need of taking the steps and several prevention programs have arisen [24]. Nevertheless, sports settings are far different from the clinical settings and labs most of the techniques were developed. Among other differences, users of a clinical service usually ask for help. They feel uneasy and request for a professional intervention. However, athletes and the rest of stakeholders are usually focused on performance and are not used to an intervention aimed at preventing an event which is not perceived as a close threat to most of them [25] and is not directly related to their sport performance. Moreover, they cannot many times be carried out in fitted rooms but in court, buses, and these interventions must be adapted to the regular sport schedules, competing with the training sessions, the tactical sessions, and the physical training. Likewise, there is a lack of injury-prevention specialized professionals working in a daily-basis with a team in many grass-root organizations. Therefore, these have helped flourishing different LI-interventions and e-Health apps.

### 3.3. LI-Interventions Focused on Physical Warmup, Training and Fitness

Among them, probably the most widely disseminated intervention is FIFA11+. This program sponsored by The Federation Internationale de Football Association [26,27], was originally developed in 2006 and consists of 15 different exercises divided in three parts: (1) running exercises at a slow speed combined with active stretching and controlled partner contacts; (2) six sets of exercises focusing on core and leg strength, balance and plyo-metrics/agility, each with three levels of increasing difficulty; and (3) running exercises at moderate/high speed combined with planting/cutting movements. Manual of the program can be downloaded directly from the website (https://www.f-marc.com/files/downloads/workbook/11plus_workbook_s.pdf, accessed on 7 December 2021) and it is expected the coaches to promote such training in their athletes’ ordinary routines.

Different studies have tested this combination of plyometric exercises, core and dynamic stabilization, eccentric thigh muscle and proprioceptive training, in all cases paying full attention to correct posture and body control. Results showed the efficacy of FIFA11+ in reducing SI risk [28,29], up to 39% [30]. Moreover, some pieces of research indicated a significant decrease in the number of injured players the teams which routinely introduced the FIFA11+ presented, ranging from 30% to 70% less than those which not [31,32]. Bizzini and Dvorak [24] not just only highlight the efficacy of the intervention different studies have reported but the general positive effects in sport performance. However, the same authors also raised several concerns regarding the implementation of the program, being more effective when the whole institution and the different bodies are involved [24].

Wilson et al. [33] have recently developed and tested the efficacy of a web-based program to increase cervical strength in adolescents. The program includes a set of exercises to be self-administered through a web site which provides with written and video recorded instructions. It consists of a six-week, three-phase, three to seven times per week session program, starting with a warmup, following with the correspondent exercises, and finishing with a cooling down task. The program does not require any specific equipment not in-person supervision by an expert professional.

Wilson et al. [33] carried out a cohort observational study comparing a group which received the intervention and a group which did not received any specific neck strength training. Results showed young athletes who received the intervention increased their cervical muscle strength after benefit from the program while the control group did not show any significant strength improvement. It is expected strengthening cervical musculature might reduce sport injury concussions and other neck and head sport injuries.

In a review, Van Mechelen et al. [34] identified up to 18 apps aimed at preventing SI, whether using neuromuscular training, tapping, warmup routines, but just only four of them were evidence-based. These four were (1) the app Ankle, which consists of a set of exercises focusing on preventing ankle sprain using neuromuscular training; (2) the app Elastoplast, which gives users advice regarding taping ankles, knees and shoulder to prevent injuries; (3) the app iPrevent ACL injuries, which consists a video demonstration of a set of exercises aimed at training warmup, stretching, strengthening, plyometrics, agility as well as a cool-down exercise; and, (4) the series of apps Medical iRehab (Anklesprain, Impingement Syndrome, Plantar Fascitis, Shoulderinestability, and Shoulder rotation cuff), all of which include some preventions tips. Even though the authors thought mobile apps may be useful in the prevention of SI, their analysis reveal the lack of scientifically sound of many of these apps.

### 3.4. LI-Interventions Centered in Mental Skills

Sport injury is a multifactorial event where many internal and external, proximal and remote factors play a role in a recursive way [35,36]. Among some of those factors whose interest in have risen in recent times, psychological variables related to athletes’ vulnerability to sports injuries devotes a special mention. The most widely accepted model linking psychological variables and SI vulnerability is Williams and Andersen’s stress–injury model [37,38]. According to this model, stressful situations the athlete has to cope with may trigger a stress response which increases athlete’s vulnerability to SI. Two mechanisms lay at the basis of the link between stress and SIs: muscle tension and attentional deficits. Stress response increases muscle tension, and that physiological response reduces flexibility, likely increases fatigue, and can disturb motor co-ordination. At the same time, stress response tends to narrow attentional field that may reduce the perception of relevant peripheral cues that can signal or prevent the occurrence of some of the events which increase the risk of sustaining a SI.

Williams and Andersen’s model has received robust empirical support [39,40] and has established managing the stress response as the core target when aiming to decrease athletes’ vulnerability to SIs using psychological interventions. Efficacy of these interventions have been tested in some systematic reviews and meta-analysis [39,41,42,43,44] which have, at least partially, supported the value of these kind of approaches in reducing SI rates and the interest in adding them to the arsenal of tools directed to reduce SI incidence.

Focused on providing a set of tools to proceed with a mental warmup before starting with the sport activity, Brewer et al. [45] developed Mental Warmup for Athletes. It consists of a recorded set of instructions which uses visual imagery to foster arousal regulation, attentional focusing, goal setting, and positive thinking. The record takes about 5 min length and usually follows some general description to athletes about what sport psychology is and its basic principles, delivered in a workshop previously organized. Authors have developed a website (http://www.supportforsport.org/mentalwarmup/index.html accessed on 7 December 2021) where athletes and coaches can download the record as well as the script to help any athlete, regardless how much sport psychology support he/she counts for.

Mental Warmup for Athletes have shown to increase psychological readiness to athletic performance after receiving the intervention and to reduce stress [45,46]. Brewer et al. [45] also reported a good level of the mental warmup intervention acceptability during the season.

Inspired by the Brewer et al. [45], Mental Warmup for Athletes, UAMSportPyschapp [47] was designed as a LI CBT intervention for young athletes aimed at managing stress response. UAMSportPsychapp uses a smartphone app that has two profiles available: player and coach. UAMSportPsychapp has been designed for iOS and Android stores that athletes can download and adjust according to their time schedule, training rhythm, and needs.

The coach profile brings coaches the opportunity to monitor athletes’ implementation of the different tasks the program includes. Therefore, coaches can act as non-specialized supportive agents overseeing their pupils’ comply with the program. It also offers the coach the possibility of assessing training session satisfaction and to compare to the athletes’ global satisfaction assessment.

The player profile offers a training in arousal control and attentional focusing using both progressive muscle relaxation and mindfulness techniques. These techniques are delivered in three phases: (1) training-phase, (2) consolidation-phase, (3) follow-up-phase. In a qualitative study among coaches who had used and monitored pupils’ use of the app, results showed very good ratings from their perspective on the accessibility, acceptance, and usefulness of the app [47]. Moreover, comparing young soccer players’ pre-post values, this app has shown high efficacy in reducing athletes’ perceived tension as well as somatic and cognitive anxiety, increasing self-confidence, these three measured by the CSAI-2R (competitive state anxiety inventory) [48]. It also has shown an increase in perceived attentional focusing and an increase in MIS (mindfulness inventory for sport) [49] awareness and refocusing scores among young soccer players [50].

## 4. How to Gamify LI interventions’ e-Health Apps to Prevent SI

The main challenge that kind on interventions has to deal with is the high levels of attrition usually associated with e-Health technology. As was mentioned, gamification may become a promising approach to increase user engagement with e-Health apps [3]. Nevertheless, gamification is more than just designing a game inside the app. According to Hunicke, LeBlanc and Zubek’s [51] mechanics–dynamics–aesthetics (MDA) framework, game has three main components: rules, system, and “fun”, biunivocally related to the design components: mechanics, dynamics, and aesthetics. Mechanics regard to those elements which set up the game. These include game rules, the control mechanisms as well as the specific components (tools, points, levels, leaderboard, etc.). Using Robson et al. [52] analogy with organizational control theory, “mechanics equate to the organizational systems and technologies that managers can use to induce the required behaviors and outcomes” (p. 414).

Dynamics describes the user’s behavior with the mechanics that generates the inputs and outputs of the participants over time. Opposite to the mechanics of the gamified experience which are set up by the designer, dynamics emerge from how the participant interacts with the app. Components of the dynamic are rewards, status, achievement, competition, etc., that promote and echoes users’ behavior [51].

Aesthetics describes the emotional responses the game intends to elicit in the player when interacting with the dynamics and regards to what makes the game enjoyable. Aesthetics are the first elements that are perceived by the user and are key in keeping the participant involved. Aesthetics produce the participant’s experience with the app including but not limited to sensation, fantasy, narrative, challenge, fellowship, discovery expression and submission [51].

### 4.1. Gamifying LI Interventions Aimed at Preventing SIs

Interventions such as the UAMSportPsychapp [47], a LI CBT program centered on muscle tension and attentional focusing as key facets related to the vulnerability to SIs, might become a useful tool. Such LI interventions allow to reach as many young athletes as possible, regardless the lack of in-site sport psychologist and some other conditions that may prevent teams and coaches to train their pupils on the basic mental skills associated to the vulnerability to sustain sport injuries. In fact, UAMSportPsychapp has shown to be a helpful instrument. As was mentioned, results showed satisfactory coaches’ appraisal about accessibility, acceptance, and usefulness of the app. They also showed to reduce perceived tension and to increase focusing. Nevertheless, users also reported low levels of engagement and labeled the app as “boring” [50], and the analysis of the attrition showed quite a high rate, which may compromise the effectiveness of the intervention to prevent SIs. Efficacy of the components of the program included in the app must come together the fun an enjoyment in doing the exercises and reaching the goals. Otherwise, effectiveness of the intervention will plunge to being ineffective.

Keeping in mind that the core difference between a videogame and a gamified intervention is that the latest is aimed at improving or fostering specific knowledge and skills, beyond those directly involved in the game, and transfer them to daily life settings, such knowledge and skills should be the framework the previous mentioned aspects should revolve around. This is to say that visual design, narrative, incentive system, spoken text and musical score, if any, are at the service of the learning goals pursued. 

Therefore, the following is a proposal to gamify phone-based LI interventions such as the UAMSportPyschapp. The proposal is focused in the three components: mechanics, dynamics, and aesthetics.

### 4.2. The Musts a Gamified Phone-Based Intervention Should Consider

Usually, interventions such as UAMSportPsychapp are merely a set of different training techniques which do not blend each other, and there is a lack of a unique narrative linking the exercises. Likewise, the activities requested are usually limited to the direct interaction with the app. Nevertheless, transfer of learning is key. Transfer depends on both, what Salomon and Perkins [53] called “low-road”: practicing extensively and variedly to promote automaticity, and what they called “high-road”: abstracting the basic elements and consciously applying them to new settings. The app should also link the exercises with real life situations athletes have to face. Similarly, feedback is crucial in guiding the learning process as well as in promoting engagement. Feedback must be immediate to performance, be adapted to the result of the daily training, and give the opportunity to rehearse the training when performance it not as good as expected. Nevertheless, the kind of skills that are trained in interventions such as these makes the feedback mostly lying on general subjective appraisals of participants. This could limit the efficacy of the feedback to promote skills’ learning. Thus, using some other outcomes may help provide more useful feedback. 

Another key element in motivating users to continue using the gamifies task is the incentive system. Regardless the task uses intrinsic-to-the-game (included in the game—for instance, reaching some tools that can be used in the following phases of the game) or extrinsic-to-the-game (not included as a part of the game but available to use in a metagame—for instance, the final score that feeds a leaderboard with the rest of the users) incentives, such elements are aimed at boosting, goal-directing or changing the users’ behavior and should be design according to a reinforcement pattern. 

As was mentioned adaptivity regards to the personalization of the game to the specific user conditions/level/needs. Adaptivity is key in promoting users’ engagement [6]. It is not easy to transform LI interventions in personalized programs when there is not a professional leading the process. Nonetheless, the intervention must feature ways in which the participant specific needs and/or (lack of) progress are considered. Gamification may help adaptivity of the intervention using closing loops and frequent assessments.

## 5. Conclusions 

Despite of the pervasiveness of sports injuries, regardless ages, competitive levels and sports, the universal generalization of preventive measures remains challenging. This lack of generalized implementation of prevention actions contrasts with the current evidence-based knowledge about effective interventions, whether focused on physical [29,54] or psychological variables [42,44]. The same applies to children and youngsters where sport is the leading cause of injuries [55] and to whom several interventions have shown to be effective [56].

Different factors relate to the lack of an extensive and faithful-to-the-protocol implementation of interventions aimed at preventing SIs. As Finch [57] quoted, interventions must be acceptable, adopted, and compiled with not just only by the athletes, but also the coaches, the medical and physical staff, and the administrators. Likewise, prevention of children and youngsters’ SI should involve different stakeholders, from the athlete to the parents, from the coach to the sport organization, from the community to the government [56].

One of the barriers a systematic implementation of preventing measures has to deal with is the lack of specialized professionals in many grass-root organizations. This is particularly true when referring to sport psychologists working with the team on a regular basis but can make it extensive to kinesiologists and fitness trainers. The need to overcome the limitations regarding the absence of highly qualified professionals in this and other fields have led to the development of what is called low-intensity interventions. These self-administered interventions, many of them based on web- and/or smartphone apps materials and with light or no specialist support and/or monitoring have broadly increased the access to interventions that, otherwise, individuals would not have received. Nevertheless, these LI-interventions are not without some cons and challenges. Probably the more important is keeping users engage in the tasks the program provides and in the use of the device the delivering is based on.

Gamification has emerged as a way to engage people in learning tasks and is currently proposed as an effective component of e-Health and other learning-oriented apps. Gamified interventions focused on learning skills and promoting behavioral change may contribute to overcome the difficulties LI-interventions have shown.

Gamified phone-based interventions must consider several aspects in order to adapt components of games to the learning goals pursued. According to these, several guidelines are proposed:

Guideline 1: We propose developing a general narrative, adapted to young athletes, comprising the diverse techniques the app is going to use and put them in relation with daily life settings. As an example, the intervention may present the task as a journey the young athlete start to become more resilient, better player. This story telling must appear each time the user logs in and should provide the framework of the rest of the components of the gamified experience (incentives, feedback).

Guideline 2: We suggest including opportunities to practice the skills beyond the exercises proposed by the intervention and being able to compute such training experiences in the app. For instance, the app may allow the user programing reminders and also asking when logging in whether the athlete has put in practice the skills and present them as a part of the feedback and collect rewards by doing so.

Guideline 3: We recommend using general subjective appraisals but also more specific outcomes (e.g., heart rate before and after the training) to provide feedback. Using other devices such as pulsimeters or smartwatches to collect these specific outcomes could be helpful. Likewise, showing the progress bar of the whole training increases competence and improves motivation.

Guideline 4: We advise to combine both intrinsic to the game and extrinsic to the game incentives. In this vein, using points or stars to let the participant move to the next level, as well awarding with badges the higher the achievement in the skills is, can help affective, behavioral, and cognitive engagement. Using leaderboards combining either the use and/or the achievement of the rest of the teammates contributes to the sociocultural engagement.

Guideline 5: We suggest creating recursive loops to allow those users who are not reaching the goals to continue practicing. Moreover, the phone-based intervention must give users who face specific challenges the skills trained can help to cope with to return back to previous steps of training that can be useful for that.

The incorporation of such approach may contribute to the design and implementation of effective prevention measures to reduce the SI incidence rate that has become a real public health concern.

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
