# Peer review of "Gamifying App-Based Low-Intensity Psychological Interventions to Prevent Sports Injuries in Young Athletes: A Review and Some Guidelines"

_ijerph, 2021, doi:10.3390/ijerph182412997_

Round 1

Reviewer 1 Report

First and foremost, I would like to extend my sincere congratulations to the authors for their work. The resulting paper shed light to some very interesting and enriching topics, even though in my opinion, it´s necessary review this article in order to improve it.

Personally, I consider that, in the introduction, more specific data from the articles quoted would be of great use.

Moreover, it’s difficult for me read and comprehend this introduction and the rest od the article due to the large quantity information. I consider that it would be interesting to reorganize all information and select the most important information. It’s necessary improve this part of the article to render the text easier to read.

Other mistake is that authors no showed what is the meaning of (CI) in point 2. Epidemiology of sport injuries in order to clarify the content of the text.

Furthermore, several points are explained but I consider that it’s necessary enhance the connection among paragraphs.

Further, I believe that would be interesting if authors decide take a look in the references section. In my opinion there are excessive bibliography (104 references) and besides, some of these references are outdated. I think that references from 1984 to 2004 should be updated. There are plenty of information about gamifying nowadays.

Broadly speaking, I think that the article can be improved and I will be delighted to reread this research.

All in all, this research is of high scientific interest for the field of gamifying and sport injuries and could provide interesting information for future studies.

Author Response

We would like to thank the Reviewer for his/her kind words regarding our work and also for taking the trouble of carrying out the review which, undoubtedly, has contributed to improve the ms. Bellow you will find how deal with your suggestions.

1.- The manuscript has been reduced in length, streamlined and also reorganized in order to facilitate a smoother reading of the text. In the current version we start with gamification as a promising approach to increase user engagement with e-Health apps. Then, we move forward to LI-interventions and what these mean. Afterwards, we talk about LI-interventions and e-Health targeting SIs. Doing so we review the current figures of SI among youngsters as well as the LI-interventions and apps designed to prevent SI focused on either physical warmup, training and fitness or on mental skills. These give way to explain how to gamify the interventions according to the Hunicke et al’s Mechanics-Dynamics-Aesthetics framework. The ms. wraps up giving some guidelines when gamifying health apps.

2.- It has been stated what CI (confidence interval) stands for when was firstly cited

3.- We have revamped the whole ms. We expect a better connection among paragraphs has been introduced.

4.- We have substantially reduced the number of citation from 104 to 57. However, we still maintained some referring to classical pieces of work, such as Salomon et al’s (1989) seminal work on learning transfer or Andersen & Williams’s (Andersen & Williams, 1988, Williams & Andersen, 1998) leading model about psychological factors making athletes more vulnerable to SIs.

5.- We hope the Reviewer considers we have improved the ms. Thank you again for your comments.

Reviewer 2 Report

This review aimed at evaluating the effects of gamifying app-based low-intensity psychological interventions to prevent sports injuries in young athletes. Although it could be interesting, in my opinion the Authors failed in building up the review as it seems not being in line with the declared aim. While going with the reading through the different section no information are provided about the effectiveness of these kind of app in reducing the risk of sport injuries (SIs)

The Authors correctly started by introducing the epidemiology of SIs and how they can be influenced by the psychological status of the athlete, providing the potential mechanisms underpinning SI onset and psychological status. From here on one can expect to found a revision of the literature linking these two aspects, but this is not: only section 5.1 is possibly centred to the review aim.

As said above, the review could also be interesting and, personally, I could be on the whole in agreement with the Authors conclusion, but I found on the whole the review of of scope.

Author Response

We would like to thank the Reviewer for his/her accurate comments. We have taken into account in order to improve the ms. Bellow you will find how deal with your suggestions.

1..- Reviewer is right the paper does not fit with a systematic review that could have been deduced from the title of the article. Nonetheless, we did not try to produce a systematic review of the effectiveness of the current apps that are aimed at preventing SIs. Our goal was to identify those which are either theory or empirically sound as well as highlighting how to improve the main concern they have to face: increase engagement and decrease attrition. Therefore, we have modified the title in order to emphasize what the ms. is focused on.

2.- We have revamped the whole paper in order to better fit the aim, chiefly focused on how LI-interventions and e-Health apps can increase engagement.

Reviewer 3 Report

- The authors present a rigorous, original, relevant and pertinent work within the field of sports injury psychology.

- It is necessary to highlight the extensive and thorough literature review that supports the relevance of "Gamification of low-intensity psychological interventions based on apps for the prevention of sports injuries in young athletes".

- As a proposal, consider the effectiveness of other mobile applications that use gamification in the educational context and that aim to help students acquire knowledge about different subjects of the curriculum or improve their linguistic competence in a foreign language.

- I consider it necessary to add a guideline on the promotion of this tool in the different organizations where the target population is integrated.

Author Response

We kindly thank the Reviewer for his/her kind words regarding our work and for taking the trouble of carrying out the review. Regarding your comments:

1.- Thank you for prasing the review of the different LI-interventions based on e-Health apps. We considered both, interventions aimed at improving physical warmup, training and fitness, and interventions aimed at improve mental skills due to both aspects should be considered when preventing SIs.

2.- We are perfectly aware of the use of gamification in different settings, such as the educational. However, in favor of the simplicity and streamline of the ms., and according to what other reviewers posed, we have decided to focus on those aimed at preventing Sis.

3.- The Reviewer is quite right when highlights the importance of implementing these kind of interventions in grass-root organization. This means there should be not just only effective but also acceptable interventions in order to cope with the huge health concern SI has become. Nevertheless, tackling the implementation issue of prevention interventions might probably demand another contribution. We have addressed this in a previous paper (Rubio, V.J.; Olmedilla, A.; Turbay, F. Low-intensitiy Cognitive-Behavioral Therapy. A mobile phone-based intervention for preventing sport injuries in football teams. In A. Ivarsson; U. Johnson (Eds.), Psychological Basis of Sport Injuries (4th Ed.) (pp. 63-769). Morgantown, WV: FIT. 2020). In any case, implementation is related to different factors, one of them being then intervention itself (how long it takes, how many resources demands,…). Since this ms. was aimed to give general guidelines when designing interventions, further pieces of work should more specifically address the implementation of the different kind of interventions.

Round 2

Reviewer 2 Report

The Authors did a big effort in replying my previous issues.  I think the present ms version could be suitable for publication.